# Testing microbiome associations with survival times at both the community and individual taxon levels

**Yingtian Hu**[1], **Yunxiao Li**[1], **Glen A. Satten**[2], **Yi-Juan Hu**[1]*

**1** Department of Biostatistics and Bioinformatics, Emory University, Atlanta, Georgia, United States of America, **2** Department of Gynecology and Obstetrics, Emory University School of Medicine, Atlanta, Georgia, United States of America

* yijuan.hu@emory.edu

## Abstract

### Background

Finding microbiome associations with possibly censored survival times is an important problem, especially as specific taxa could serve as biomarkers for disease prognosis or as targets for therapeutic interventions. The two existing methods for survival outcomes, MiR-KAT-S and OMiSA, are restricted to testing associations at the community level and do not provide results at the individual taxon level. An ad hoc approach testing each taxon with a survival outcome using the Cox proportional hazard model may not perform well in the microbiome setting with sparse count data and small sample sizes.

### Methods

We have previously developed the linear decomposition model (LDM) for testing continuous or discrete outcomes that unifies community-level and taxon-level tests into one framework. Here we extend the LDM to test survival outcomes. We propose to use the Martingale residuals or the deviance residuals obtained from the Cox model as continuous covariates in the LDM. We further construct tests that combine the results of analyzing each set of residuals separately. Finally, we extend PERMANOVA, the most commonly used distance-based method for testing community-level hypotheses, to handle survival outcomes in a similar manner.

### Results

Using simulated data, we showed that the LDM-based tests preserved the false discovery rate for testing individual taxa and had good sensitivity. The LDM-based community-level tests and PERMANOVA-based tests had comparable or better power than MiRKAT-S and OMiSA. An analysis of data on the association of the gut microbiome and the time to acute graft-versus-host disease revealed several dozen associated taxa that would not have been achievable by any community-level test, as well as improved community-level tests by the LDM and PERMANOVA over those obtained using MiRKAT-S and OMiSA.

**Data Availability Statement:** The new methods described here have been added to our R package LDM, which is available on GitHub at https://github.com/yijuanhu/LDM.

**Funding:** This research was supported by the National Institutes of Health awards R01GM116065 (YJH, GAS) and R01GM141074 (YJH, GAS). The funders had no role in study design, data collection and analysis, decision to publish, or preparation of the manuscript.

**Competing interests:** The authors have declared that no competing interests exist.

## Conclusions

Unlike existing methods, our new methods are capable of discovering individual taxa that are associated with survival times, which could be of important use in clinical settings.

## Author summary

High-throughput sequencing of 16S gene or metagenomes provides an unprecedented opportunity to discover microbial associations with traits such as clinical outcomes or environmental factors. Detecting individual taxa associated with survival times has significant implications: the taxa could serve as biomarkers for disease prognosis or as targets for therapeutic interventions. However, the taxon data are highly complex because they are high-dimensional, sparse (having 50–90% zero counts), and highly overdispersed. Existing methods for microbial associations with survival outcomes are restricted to testing associations at the community level and do not provide results at the individual taxon level. An ad hoc approach testing each taxon with a survival outcome using the Cox proportional hazard model may not perform well in the microbiome setting with sparse count data. We present an approach that can be used by the LDM and PERMANOVA for testing microbial associations with survival outcomes at both the community and individual taxon levels. In particular, we provide the first test at the individual taxon level. Therefore, our work represents a major advance in analytical methods for microbial association studies and will have a strong impact on current and future microbiome research in clinical settings.

This is a *PLOS Computational Biology* Software paper.

## Introduction

Advances in sequencing technologies for profiling human microbiomes have led to the discovery of numerous microbiome associations with clinical responses [1–3]. These successes suggest that microbial taxa may be useful as biomarkers for disease prognosis, or targets for therapeutic interventions [4]. For example, the miCARE study is attempting to find whether the gut microbiome can be used to predict colorectal cancer recurrence (Principal Investigator: Dr. Veronika Fedirko, personal communication). Like the miCARE study, studies conducted to establish these links would collect the subjects' times to an event of interest (i.e., survival times) as the outcomes, some of which may have censored values. For the success of this research, finding microbiome associations with the survival outcomes only at the community level may be less important than finding associations with individual taxa (we use "taxon" generically to refer to any feature such as amplicon sequence variants or any other taxonomic or functional grouping of bacterial sequences).

However, data from microbiome association studies can be difficult to analyze, because the taxa count data may have hundreds to thousands of taxa and 50–90% zero counts, and are typically highly overdispersed. In addition, there generally exist confounders, such as previous treatment history or current medications, that correlate with both the microbiome composition and the survival outcome and so must be properly adjusted for. Finally, the sample size in

a microbiome association study is typically not large and the event rate may be low, especially for rare diseases such as cancers. Analysis methods that cannot account for these data complexities will typically not yield robust and clinically meaningful results.

Two methods have been proposed specifically for testing association between the microbiome and survival outcomes: MiRKAT-S [5] and OMiSA [6]. Unfortunately, both methods are restricted to community-level (global) association tests. While OMiSA does allow testing pre-determined sets of taxa such as taxonomic classes, it requires each set to be comprised of multiple taxa. As a result, neither MiRKAT-S nor OMiSA can be used to find individual taxa that can act as biomarkers. A third, ad hoc, approach is to apply the Cox proportional hazard model [7] in a taxon-by-taxon manner [8, 9]. However, the performance of this approach has not been formally evaluated in the microbiome context, although it is known that small sample sizes and sparse count data may lead to inflated type I error when using the Cox model [10, 11]. Unfortunately, permutation-based inference, which might improve the performance of the ad hoc approach, is difficult for survival outcomes.

We previously proposed the linear decomposition model (LDM) [12] for testing microbiome associations with continuous or categorical (including binary) outcomes, which not only performs the test at the community level but also at the individual taxon level with false discovery rate (FDR) control. Here, we extend the LDM to survival outcomes, in order to allow a unified test framework to test both community-level and taxon-specific associations for survival outcomes. The LDM is based on a linear model that regresses the microbial data at each taxon on the (confounding) covariates that we wish to adjust for and the outcome variable(s) that we wish to test. Inference is based on permutation to circumvent making parametric assumptions about the distribution of the taxon-level data. In addition, the LDM is highly versatile: it can analyze the taxon-level data at the relative abundance scale, the arcsin-root-transformed relative abundance scale (which is variance-stabilizing for Multinomial and Dirichlet-Multinomial count data) or any other transformation, as well as the presence-absence scale [13], and can also accommodate clustered samples [12, 14].

Our extension of the LDM was motivated by ideas developed in MiRKAT-S and OMiSA. Both of these tests first fit a Cox model to account for the relationship between any fixed covariates (excluding microbiome variables) and survival times. Then, using a random-effects framework, the variance-covariance matrix of the (Martingale) residuals from the Cox model are compared to a between-sample distance matrix calculated using the microbiome data; the similarity between these two matrices indicates the extent of association between the microbiome and the survival outcome. MiRKAT-S allows an arbitrary distance matrix, most commonly, the Bray-Curtis or Jaccard distance matrix. OMiSA extends MiRKAT-S by using a family of power transformations of the relative abundance data to weigh abundant and rare taxa differently but calculating the MiRKAT-S test statistic based on the Euclidean distance only. Our generalization of the LDM to survival outcomes is also based on obtaining residuals from the Cox model; however, we use these residuals as covariates in the LDM to directly assess the association between the microbiome and the survival outcome. In this way, we are able to use the LDM to test both community-level and taxon-level associations with a survival outcome. In a similar manner, we also extend PERMANOVA [15], the most commonly used method for testing microbiome associations, to handle survival outcomes, although the test is at the community level and distance-based like MiRKAT-S.

The rest of this paper is organized as follows. In the Methods section, we first describe our tests based on the Martingale residuals, showing their connection to MiRKAT-S, OMiSA, and the taxon-by-taxon Cox regression. Then we extend the tests to use the *deviance* residuals, which are transformations of the Martingale residuals that are more symmetric above zero, and then construct *combination* tests that combine the results from tests using the two types of

residuals. In the Results section, we first present simulation studies and then an application of all methods to data on acute graft-versus-host disease (aGVHD) after allogeneic blood or marrow transplantation [16]. We conclude with a brief discussion section.

## Methods

Suppose that, for $n$ unrelated subjects, we have data on the time to an event of interest (e.g., disease onset or relapse) that may be subject to random censoring. For $i = 1, 2, \ldots, n$, let $T_i$ be the (underlying) time to event for the $i$th subject and $C_i$ be the corresponding censoring time. Instead of observing $T_i$ and $C_i$, we only observe the time $U_i = \min(T_i, C_i)$ and the indicator $\Delta_i = I(T_i \leq C_i)$ that indicates whether $U_i$ corresponds to the event or to censoring. Further, let $X_i$ be a set of possibly confounding covariates, which does not include the intercept. For $j = 1, 2, \ldots, J$, let $Z_{ij}$ denote the microbiome data on taxon $j$ from subject $i$, which can be the relative abundance, arcsin-root-transformed relative abundance, presence-absence status, or any (e.g., additive or centered) log-ratio transformed data. Following the conventions used in the LDM, we assume that both $X_i$ and $Z_{ij}$ are centered to have mean zero, i.e., $\sum_{i=1}^{n} X_i = 0$ and $\sum_{i=1}^{n} Z_{ij} = 0$ for any $j$.

Because survival times are censored, it is difficult to include them in the linear model framework used by the LDM. Following MiRKAT-S [5], we resolve this issue by first fitting a Cox model to the survival outcomes $(U_i, \Delta_i)$ and covariate data $X_i$; we then use the residuals from this model as a covariate in the LDM [12]. Because no microbiome data is used in the Cox model, the residuals should be associated with the microbiome data if the microbiome affects the survival outcome. If we use the Martingale residuals, denoted by $M_i$ for subject $i$, we propose to test the association of taxon $j$ with the Martingale residuals while adjusting for covariates $X_i$ by using the LDM to fit the following linear model:

$$Z_{ij} = \beta_{X,j}^{\mathrm{T}} X_i + \beta_j M_i + e_i, \tag{1}$$

where $e_i$ is the error term with mean zero and a constant variance (the only distributional assumption we make). Note that the Martingale residuals have the properties that $\sum_{i=1}^{n} M_i = 0$ and $\sum_{i=1}^{n} M_i X_i = 0$ [11].

To test $H_0 : \beta_j = 0$, the LDM uses an $F$-statistic, the numerator of which is proportional to the square of $\mathbb{U}_j$ given by

$$\mathbb{U}_j = \sum_{i=1}^{n} M_i(Z_{ij} - \hat{\beta}_{X,j}^{\mathrm{T}} X_i) = \sum_{i=1}^{n} M_i Z_{ij},$$

where $\hat{\beta}_{X,j}$ is the least squares estimator of $\beta_{X,j}$ under the null model of (1). Further, the numerator of the global test statistic for testing the global association between the Martingale residuals and the microbiome is

$$\mathbb{U}_{\mathrm{global}}^2 = \sum_{j=1}^{J} \mathbb{U}_j^2.$$

These test statistics can be used to show a connection between our approach and existing methods. First, the global statistic $\mathbb{U}_{\mathrm{global}}^2$ agrees with the variance-component score statistic in MiRKAT-S when the Euclidean distance (the linear kernel) is used, as well as the variance-component score statistic in OMiSA (the OMiSALN part) for untransformed data. Second, letting $\lambda(\cdot)$ denote the hazard function for a survival analysis, the taxon-specific $\mathbb{U}_j$ coincides with the score statistic for testing $\alpha_j = 0$ in the Cox model $\lambda(t; X_i, Z_{ij}) = \lambda_0(t)\exp(\alpha_X^{\mathrm{T}} X_i + \alpha_j Z_{ij})$ [11],

which includes both the covariates and the microbiome data from the $j$th taxon as explanatory variables in the hazard function. These connections justify the use of the Martingale residual as a covariate in the LDM.

The main advantage of our approach is that results for individual taxa are available, and that the global test statistic is a coherent combination of these taxon-specific statistics; neither MiRKAT-S nor OMiSA provide taxon-specific results. However, the LDM is based on the Euclidean distance for combining taxon-specific statistics, while MiRKAT-S can use arbitrary distances. For this reason, we also provide an extension of PERMANOVA for testing survival outcomes that can be used with arbitrary distances, at the end of this section.

An important feature of our approach is that, although the effect of $X_i$ has been removed from $M_i$ (i.e., $M_i$ and $X_i$ are uncorrelated), we still include $X_i$ in (1). In the S1 Text, we show how including this term allows our permutation tests to achieve higher power than the permutation tests currently available in MiRKAT-S. We further show how to obtain global tests with power similar to what we achieve using the original MiRKAT method [17] with the Martingale residual as a continuous outcome.

Compared with the ad hoc approach of fitting a Cox model for each taxon, our permutation-based inference is robust to small sample size, low event rate, and sparse count data, while the Cox model is known to have inflated type I error in these situations [10, 11]. Compared with the ad hoc approach, both MiRKAT-S and our approach share the huge computational advantage that the Cox model only needs to be fit once. In addition, both methods only depend on the presence of an association between the Martingale residuals and the microbiome measures, and so do not depend on the correct specification of the Cox model for validity (i.e., type I error control), although power may be lost if the Cox model provides a poor fit to the data.

One deficiency of the Martingale residual is its skewness, because it has a maximum value 1 but a minimum value $-\infty$. Because a residual measure with a more normal-like distribution may perform better in downstream analyses, Therneau et al. [18] introduced the deviance residual for a Cox model:

$$D_i = \text{sign}(M_i)\sqrt{-2\{M_i + \Delta_i \log(\Delta_i - M_i)\}},$$

which is a non-linear transformation of the Martingale residual $M_i$. Therneau et al. found that with less than 25% censoring, the deviance residual is approximately normally distributed; with more than 40% censoring, too many points will lie near 0 making the distribution non-normal, although the deviance residuals remain approximately symmetric about 0. Therefore, we also consider a variation of our method that replaces $M_i$ by $D_i$ in the linear model (1). Although $D_i$ is not orthogonal to $X_i$, we can still use the LDM to fit (1) as long as $X_i$ enters the model before $D_i$ because, in this case, the LDM will make $D_i$ orthogonal to $X_i$ before testing for association with $Z_{ij}$. In our simulations, use of the Martingale residual sometimes gave better power and sensitivity; in other situations the deviance residual performed better. Since we cannot characterize those scenarios a priori, we also combine the results from analyzing each residual separately into a single combination test. To account for differences in residual scale, we take the minimum of the $p$-values obtained from analyzing each residual separately, and use the corresponding minima of null $p$-values for each test from the permutation replicates to simulate the null distribution; the null $p$-value is calculated based on the rank of the test statistic among all permutation replicates [19].

We extend PERMANOVA to analyzing survival outcomes in a similar way. Like MiRKAT-S, PERMANOVA is distance-based and offers a global test of the association at the community level. To explain the variability in a given distance matrix, we use a similar linear

model as in (1) that includes the covariates $X_i$ and the Martingale residual $M_i$ as explanatory variables. We obtain the $p$-value for testing $M_i$, repeat the procedure with the deviance residual $D_i$, and then construct a combination test that take the minimum of the two $p$-values as the final test statistic. A common use of PERMANOVA is through the function "adonis2" in the R package vegan. We have also provided an alternative implementation of PERMANOVA through the function "permanovaFL" in our LDM package [12], which differs from adonis2 in the way permutations are conducted. We found that permanovaFL outperforms adonis2 in many situations [12, 14, 20].

## Results

### Simulated design

We conducted simulation studies to evaluate the properties of our approach and compare our results to those obtained using competing methods. Our simulations were based on data on 856 taxa of the upper-respiratory-tract (URT) microbiome [21] that were also used in the MiR-KAT-S paper. We considered a binary confounder $X_i$ and assumed equal numbers of subjects with $X_i = 1$ and $X_i = 0$. We randomly sampled 100 taxa to be associated with $X_i$ and generated their associations as follows. We first set two vectors, $\pi_1$ and $\pi_2$, to the taxon frequencies (i.e., relative abundances) estimated from the URT microbiome data, and then permuted the frequencies in $\pi_2$ that belong to the set of 100 taxa selected to be associated with $X_i$, which ensured the same frequencies in $\pi_1$ and $\pi_2$ for taxa not selected. Next, we defined a subject-specific frequency vector to be $\tilde{\pi}(X_i) = (1 - \beta_{XZ}X_i)\pi_1 + \beta_{XZ}X_i\pi_2$, in which $\beta_{XZ}$ can be interpreted as the effect of $X_i$ on the selected taxa. When $\beta_{XZ} = 0$, there was no association between $X_i$ and the microbiome, in which case $X_i$ reduced to a simple covariate for the survival outcome instead of a confounder. Finally, we generated the taxon count data for each subject using the Dirichlet-Multinomial (DM) model with mean $\tilde{\pi}(X_i)$, overdispersion 0.02, and library size sampled from $N(10000, (10000/3)^2)$ and left-truncated at 1000.

We considered two models, M1 and M2, for simulating the survival outcome. In what follows, we number the taxa by decreasing relative abundance so that taxon 1 is the most abundant. In model M1, we assumed that the relative abundances of taxa 1–10 determined the association with the survival outcome; in model M2, we assumed that the presence or absence of 10 randomly selected taxa, selected from taxa 11–100, determined this association. Specifically, we defined $S_i = \sum_{j \in \mathcal{A}} \delta_j Z_{ij}/\bar{Z}_j$ under M1 and $S_i = \sum_{j \in \mathcal{A}} \delta_j \mathbb{I}(Z_{ij} > 0)$ under M2, where $\delta_j$s were directions taking values 1 and −1 with equal probabilities (and fixed across replicates of data), $\mathcal{A}$ was the set of selected "causal" taxa, $Z_{ij}$ was the observed frequency (taxon count divided by the library size), and $\bar{Z}_j$ was the average frequency for the $j$th taxon across subjects. Then, we simulated the time to event from the Cox model with the baseline hazard following the Weibull distribution $\mathcal{W}(2, 0.01)$, namely, $T_i = 10B_i^{-1/2}(-\log V_i)^{1/2}$, where $V_i$ was sampled from the uniform distribution $U[0, 1]$ and $B_i = \exp\{\beta_{XS}\text{scale}(X_i) + \beta\text{scale}(S_i)\}$ with $\beta$ characterizing the effects of the "causal" taxa on the event time, $\beta_{XS}$ being fixed at 0.5, and scale(.) standardizing the input vector to have mean 0 and standard deviation 1. The censoring time was simulated independently from the Exponential distribution $\text{Exp}(\mu)$, where $\mu$ was set to 0.03, 0.08, and 0.2 to achieve approximately 25%, 50%, and 75% censoring. Using this procedure, we generated $n = 100$ or 50 subjects for each replicate of data. To evaluate robustness of our methods to violation of the proportional hazard (PH) assumption, we also simulated the event time from the accelerated hazard (AH) model [22] with the baseline hazard following the log-normal distribution, namely, $T_i = B_i^{-1} \exp\{\Phi^{-1}[1 - \exp(B_i \log V_i)]\}$, where $\Phi^{-1}$ is the inverse cumulative distribution function of the standard normal distribution. The censoring time was

simulated as before using $\mu$ = 0.5 to achieve approximately 50% censoring. The AH model generated data that strongly violated the PH assumption (specifically, 28.8% rejection rate for testing the PH assumption [23] using our simulated data, which was much higher than the nominal level 5% of the test) and even had crossing survival curves.

Prior to analysis, we filtered out taxa that were found in fewer than 5 subjects in the dataset. We used the R package `Survival` to obtain the Martingale and deviance residuals, $M_i$ and $D_i$, from fitting the Cox model for the survival outcomes with $X_i$ as the explanatory covariate.

For testing individual taxa, we applied the LDM with either $X_i$ and $M_i$ as covariates or $X_i$ and $D_i$ as covariates in the linear regression model (1), and refer to them as LDM-m and LDM-d, respectively. Specifically, for data generated under model M1, we applied the LDM to the relative abundance data and arcsin-root-transformed relative abundance data separately and used the omnibus test that combined their results; for data generated under model M2, we applied the LDM to the presence-absence data. We also obtained the combination test that combines the results from LDM-m and LMD-d, and refer to it as LDM-c. To evaluate the ad hoc approach, we fit the Cox model and the Firth-corrected Cox model (using the "coxphf" function in the R package `coxphf`) taxon by taxon, using $X_i$ and the taxon relative abundance under model M1 or taxon presence-absence status under model M2 as covariates; the $p$-values for these taxon-specific tests were then adjusted for multiple testing using the Benjamini-Hochberg procedure [24]. We evaluated the sensitivity and empirical FDR at nominal level 10% for all taxon-specific tests, using 1000 replicates of data.

For testing global association, we obtained these results from LDM-m, LDM-d, and LDM-c, and we also applied permanovaFL in a similar way to obtain permanovaFL-m, permanovaFL-d, and permanovaFL-c. For permanovaFL-based tests and all other distance-based tests described below, we used the Bray-Curtis distance under model M1 and the Jaccard distance (without rarefying the taxa count table since the library sizes were balanced in the simulation) under model M2. For comparison, we applied MiRKAT-S using the permutation $p$-value, which was based on the Martingale residual only. We also applied OMiSA, specifically OMiSALN, the part of OMiSA that combines the results from analyzing differently power-transformed relative abundance data (with the default set of power values), which always analyzes data at the relative abundance scale even under model M2. In addition, we considered a number of secondary tests to gain more insights. To verify the equivalence of MiRKAT-S to an implementation of MiRKAT, we applied MiRKAT with a linear regression model that used the Martingale residual as the continuous outcome and the microbiome profile as the covariates without adjusting for $X_i$, and refer to this test as MiRKAT-m1. We also applied a variation of MiRKAT-m1 that additionally adjusted $X_i$ in the linear regression, referred to as MiRKAT-m, and a variation of MiRKAT-m that replaced the Martingale residual by the deviance residual, referred to as MiRKAT-d. Finally, we applied PERMANOVA implemented in adonis2, with either $X_i$ and $M_i$ as covariates or $X_i$ and $D_i$ as covariates to obtain adonis2-m and adonis2-d. All global tests were evaluated on their type I error and power at the nominal level 0.05, based on 10000 and 1000 replicates of data, respectively.

## Simulation results

We focus on the results from simulated data with 50% censoring and sample size 100; the results when the censoring rate was varied to 75% or 25% or the sample size was reduced to 50 showed the same patterns and are thus deferred to Supplementary Materials (S1 Table, S3–S5 Figs). Fig 1 shows the sensitivity and empirical FDR results for the taxon-specific tests. In both scenarios M1 and M2, the deviance residual (LDM-d) corresponds to higher sensitivity than the Martingale residual (LDM-m), although the difference was small. We explored two more

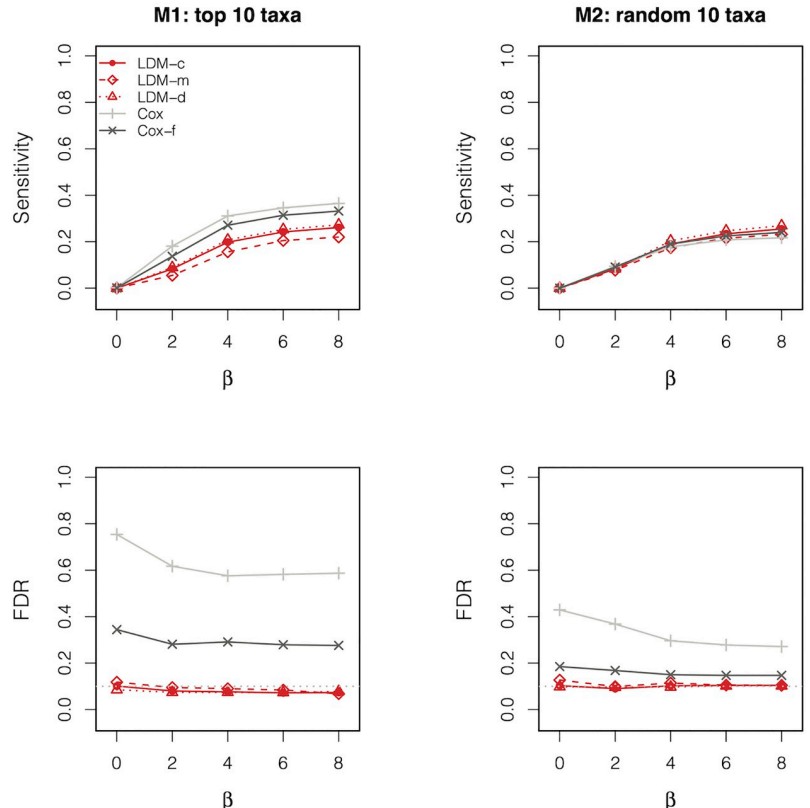

**Fig 1. Sensitivity and empirical FDR of the taxon-specific tests in analysis of simulated data with a confounder $X_i$ ($\beta_{XZ} = 0.8$), 50% censoring, and $n = 100$.** "Cox-f" is the Firth-corrected Cox model. The gray dotted line represents the nominal FDR level 10%.

scenarios, one assuming taxon 11 to be associated with the event time (referred to as M3) and one assuming taxon 21 to be associated (referred to as M4), in which data were analyzed at the relative abundance scale and the presence-absence scale, respectively. The results were displayed in S1 Fig. We found that the Martingale residual led to higher sensitivity than the deviance residual under M3 and the two residuals performed very differently under M4. Fortunately, the combination test LDM-c tracked the results of the best-performing residual in all cases. As expected, all LDM tests controlled the FDR (except for some minor inflation when the sensitivity was extremely low). The ad hoc Cox regression had very inflated FDR in all cases; the Firth-corrected Cox regression also had inflated FDR, albeit to a lesser degree. We hypothesized that the inflated FDR for both methods is due to the sparsity of the data, with zero counts for many observations at many taxa. To confirm this hypothesis, we varied the overdispersion parameter from 0.02 to 0.002 and finally to 0.0002 to successively decrease the number of cells with zero counts; results from these simulations are found in S6 Fig. Indeed, as the data became less sparse, the FDR of both Cox models became less inflated.

The type I error results of the global tests are summarized in Table 1, which shows that the LDM- and permanovaFL-related tests all yielded type I error close to the nominal level 0.05. MiRKAT-S and OMiSA produced conservative type I errors when $X_i$ was a confounder; for example, their type I error rates were 0.007 and 0.034 under model M2. Note that all these tests yielded highly inflated type I error ($> 0.4$) when the confounder was not adjusted for in the entire analysis, confirming that we have generated substantial confounding effects. The type I

**Table 1. Type I error of the global tests for simulated data with 50% censoring and $n$ = 100.**

| Hazards model | Scenario | $\beta_{XZ}$ | LDM- | | | permanovaFL- | | | MiRKAT-S | OMiSA |
|---|---|---|---|---|---|---|---|---|---|---|
| | | | c | m | d | c | m | d | | |
| Cox | M1 | 0 | 0.051 | 0.049 | 0.047 | 0.051 | 0.051 | 0.048 | 0.052 | 0.050 |
| | | 0.8 | 0.050 | 0.047 | 0.048 | 0.052 | 0.051 | 0.050 | 0.032 | 0.034 |
| | | 0.8* | 0.626 | 0.634 | 0.563 | 0.450 | 0.453 | 0.418 | 0.471 | 0.518 |
| | M2 | 0 | 0.044 | 0.042 | 0.044 | 0.046 | 0.046 | 0.044 | 0.048 | 0.050 |
| | | 0.8 | 0.048 | 0.050 | 0.047 | 0.049 | 0.050 | 0.046 | 0.007 | 0.034 |
| | | 0.8* | 0.805 | 0.808 | 0.74 | 0.814 | 0.817 | 0.74 | 0.818 | 0.518 |
| AH | M1 | 0 | 0.050 | 0.051 | 0.054 | 0.050 | 0.050 | 0.052 | 0.052 | 0.052 |
| | | 0.8 | 0.050 | 0.048 | 0.045 | 0.051 | 0.050 | 0.050 | 0.034 | 0.034 |
| | M2 | 0 | 0.050 | 0.049 | 0.049 | 0.051 | 0.052 | 0.050 | 0.053 | 0.052 |
| | | 0.8 | 0.052 | 0.050 | 0.053 | 0.050 | 0.046 | 0.051 | 0.007 | 0.034 |

Note: AH is the accelerated hazards model [22]. When $\beta_{XZ} = 0$, $X_i$ was a simple covariate (i.e., not correlated with the microbiome data); when $\beta_{XZ} = 0.8$, $X_i$ was a confounder; when $\beta_{XZ} = 0.8^*$, $X_i$ was a confounder but omitted in the entire analysis.

error rate of all these tests were robust to violation of the PH assumption when the event times were instead simulated using the AH model.

Fig 2 displays the power for the global tests. Using either the LDM or permanovaFL, the Martingale and deviance residuals, as well as the combination test, all led to similar power.

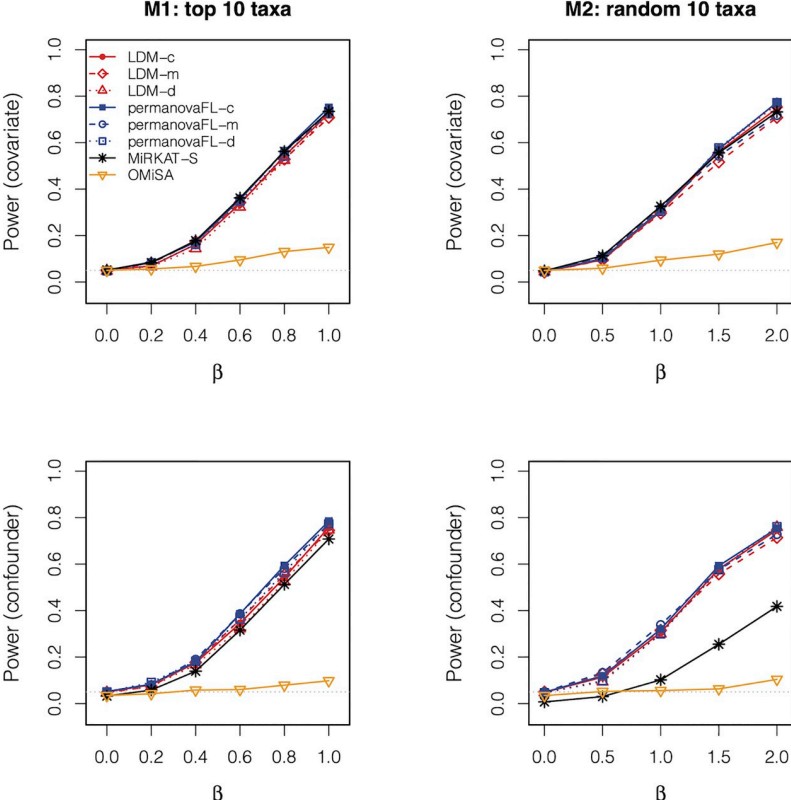

**Fig 2. Power of the global tests in the presence of a covariate ($\beta_{XZ} = 0$) and a confounder ($\beta_{XZ} = 0.8$).** The data were simulated with 50% censoring and $n$ = 100. The gray dotted line represents the nominal type I error level 0.05.

The similar power between the LDM and permanovaFL was a coincidence here and is not guaranteed in general, since permanovaFL results will vary depending on the distance measure used. MiRKAT-S had similar power to permanovaFL-m when $X_i$ was a simple covariate (i.e., not correlated with the microbiome data) but had lower power than permanovaFL-m when $X_i$ was a confounder (especially under Model M2), which is consistent with its conservative type I error results in this situation. OMiSA had very low power in both scenarios M1 and M2. We explored an additional scenario in which rare taxa (taxa 91–100) were associated with the event time; OMiSA yielded good power among all tests when the data were simulated and analyzed based on the relative abundance scale (S2 Fig).

Fig 3 displays the power for the secondary global tests and included MiRKAT-S again as a calibration. Indeed, MiRKAT-S had equivalent power to MiRKAT-m1 in all cases. MIRKAT-m and MiRKAT-d always had very similar power to permanovaFL-m and permanovaFL-d, respectively, which was expected given the equivalent performance of MiRKAT and permanovaFL we have consistently observed in the context of testing continuous or binary outcomes. These results confirmed that the improvement in the power of permanovaFL-m over MiRKAT-S was truly due to its inclusion of $X_i$ in the linear regression model (1). Lastly, adonis2-m and adonis2-d occasionally had lower power than permanovaFL-m and permanovaFL-d, as seen before [12, 14, 20].

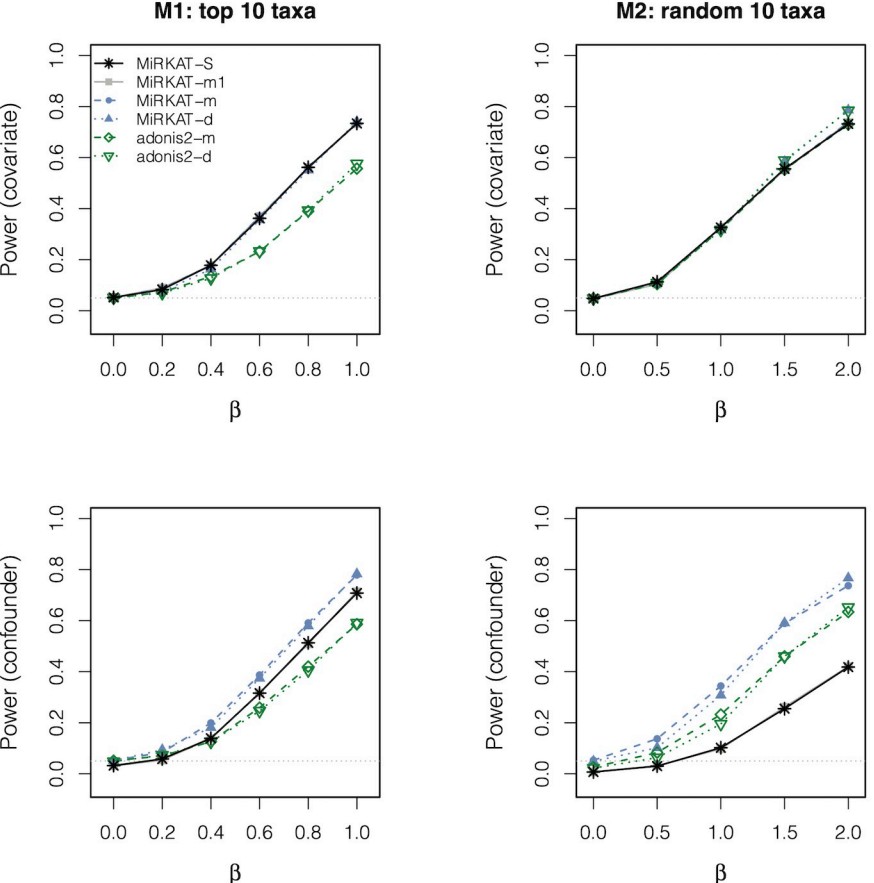

**Fig 3. See the caption to Fig 2.** The MiRKAT-S results are the same as those in Fig 2.

## Analysis of the aGVHD data

We analyzed the same data on aGVHD [16] that were also analyzed in the MiRKAT-S paper. We first followed the same procedure as in the MiRKAT-S paper to process the 16S rRNA sequencing data to obtain 2436 operational taxonomic unites (OTUs) in 94 subjects, and then removed subjects with library sizes less than 1000 and excluded OTUs that were found in fewer than 5 subjects to obtain a final set of 88 subjects and 441 OTUs for our analysis. We tested the association of the gut microbiome with two survival outcomes separately, the overall survival and the time to stage-III aGVHD, both adjusting for age and gender. The censoring rates for the overall survival and the time to stage-III aGVHD were 52.3% and 42.0%, respectively. The Martingale and deviance residuals obtained from the Cox model with age and gender as covariates were displayed in S7 Fig, which shows that neither residuals were normally or symmetrically distributed in this dataset.

We applied the LDM, the Cox model, and the Firth-corrected Cox model for testing individual OTUs, and the LDM, permanovaFL, MiRKAT-S, and OMiSA for testing the global association. We applied these methods to both relative-abundance and presence-absence data scales, in the same way as in the simulation studies; in particular, we used the OMiSALN part only for OMiSA in analysis of the relative abundance data. For the presence-absence analyses, we considered both rarefied and unrarefied data for all methods. The unrarefied data may be subject to confounding by the library size, which varied considerably between 1,274 and 265,352 in this dataset. In the rarefaction-based analysis with rarefaction depth 1,274, the LDM was based on all rarefied OTU tables (the LDM-A method in [13]), and permanovaFL and MiRKAT-S were based on the expected Jaccard distance matrix over all rarefied OTU tables [20]. Unfortunately, the Cox model and Firth-corrected Cox model cannot handle multiple rarefied OTU tables except by manually rarefying and combining the results, while OMiSALN cannot be used for presence-absence analysis.

Whenever possible, we also constructed the omnibus test for each method that combined their results from analyzing the relative abundance data and the presence-absence data (with all rarefactions). For LDM-m, LDM-d, and LDM-c, we applied LDM-omni3 [25] (an omnibus test that combines results from analyzing three data scales: relative abundance, arcsin-root-transformed relative abundance, and presence-absence scales) when analyzing the residuals of survival times. For permanovaFL-m, permanovaFL-d, and permanovaFL-c, we constructed an omnibus test based on the Bray-Curtis and Jaccard (using all rarefactions) distances. OMiSA itself is an omnibus test that combines results of OMiSALN and the omnibus version of MiRKAT-S (based on the Bray-Curtis distance and the weighted, unweighted, and generalized UniFrac [26, 27] distances without rarefaction). The Cox models and MiRKAT-S (original implementation in [5]) do not provide such omnibus tests; we did not construct a Cox model omnibus test combining relative abundance and presence-absence analyses since the marginal performance of the Cox models was so poor.

All test results were summarized in Table 2. The LDM or permanovaFL combination tests (LDM-c, permanovaFL-c) always tracked the better results obtained using the Martingale residual and the deviance residual, so we focus on their combination tests hereafter. Among the different analyses we performed, presence-absence analyses based on all rarefied OTU tables consistently led to the most significant results for all tests. Specifically, LDM-c detected 17 OTUs associated with the overall survival and 29 OTUs associated with the time to stage-III aGVHD; the survival functions stratified by the presence and absence status of each detected OTU (based on a singly rarefied OTU table) were plotted in Figs 4 and 5, which showed a clear separation in each case. LDM-c, permanovaFL-c, and MiRKAT-S yielded $p$-values 0.0002, 0.0006, and 0, respectively, for testing the global association of the gut microbiome with the

**Table 2. Results in analysis of the aGVHD data.**

| | | | Relative abundance | Presence-absence (unrarefied) | Presence-absence (all rarefactions) | Omnibus test |
|---|---|---|---|---|---|---|
| Overall survival | Number of detected OTUs | LDM-c | 2 | 2 | 17 | 3 |
| | | LDM-m | 5 | 10 | 28 | 16 |
| | | LDM-d | 0 | 1 | 3 | 1 |
| | | Cox | 0 | 2 | - | - |
| | | Cox-f | 0 | 2 | - | - |
| | Global *p*-value | LDM-c | 0.0640 | 0.0456 | 0.000200 | 0.000200 |
| | | LDM-m | 0.0565 | 0.0385 | 0.000200 | 0.000200 |
| | | LDM-d | 0.0965 | 0.0737 | 0.000400 | 0.00100 |
| | | permanovaFL-c | 0.0785 | 0.0376 | 0.000600 | 0.000800 |
| | | permanovaFL-m | 0.0665 | 0.0316 | 0.000800 | 0.00140 |
| | | permanovaFL-d | 0.132 | 0.0411 | 0.000400 | 0.000500 |
| | | MiRKAT-S | 0.0581 | 0.0290 | 0 | - |
| | | OMiSA | 0.002 | - | - | 0.01 |
| Time to stage-III aGVHD | Number of detected OTUs | LDM-c | 12 | 12 | 29 | 29 |
| | | LDM-m | 50 | 15 | 64 | 57 |
| | | LDM-d | 0 | 8 | 8 | 4 |
| | | Cox | 0 | 0 | - | - |
| | | Cox-f | 0 | 5 | - | - |
| | Global *p*-value | LDM-c | 0.0376 | 0.0365 | 0.000600 | 0.00180 |
| | | LDM-m | 0.0315 | 0.0323 | 0.000600 | 0.00160 |
| | | LDM-d | 0.0591 | 0.0668 | 0.00180 | 0.00400 |
| | | permanovaFL-c | 0.0366 | 0.0411 | 0.00160 | 0.00300 |
| | | permanovaFL-m | 0.0310 | 0.0355 | 0.00140 | 0.00260 |
| | | permanovaFL-d | 0.0604 | 0.0624 | 0.00260 | 0.00500 |
| | | MiRKAT-S | 0.0711 | 0.0300 | 0.00100 | - |
| | | OMiSA | 0.004 | - | - | 0.012 |

Note: The OTUs were detected by controlling the FDR at 10% level. The permanovaFL and MiRKAT-S results were based on the Bray-Curtis distance in analysis of relative abundance data and the Jaccard distance in analysis of presence-absence data. The omnibus tests for LDM combined results from analyzing the relative abundance, arcsin-root transformed relative abundance, and presence-absence (all rarefactions) data. The omnibus tests for permanovaFL combined results from the relative abundance scale (using the Bray-Curtis distance) and the presence-absence scale (using the Jaccard distance and averaging over all rarefactions).

overall survival, and 0.0006, 0.0016, and 0.001 for the global association with the time to stage-III aGVHD. The substantial difference in results between the rarefied and unrarefied analyses implied that differences in the library size played an important, although undesired, role in the unrarefied analysis. Based on the relative abundance data and a nominal significance level 0.05, LDM-c and permanovaFL-c declared a significant global association of the gut microbiome with the time to stage-III aGVHD but failed for the overall survival; MiRKAT-S failed for both outcomes; OMiSA was significant for both outcomes. The omnibus test results tracked the results of the best-performing data scale in all cases.

## Discussion

We have presented an approach that can be used in the LDM and PERMANOVA frameworks to testing microbiome associations with survival outcomes. This approach is based on a linear model treating both the Martingale and deviance residuals from the Cox proportional hazards

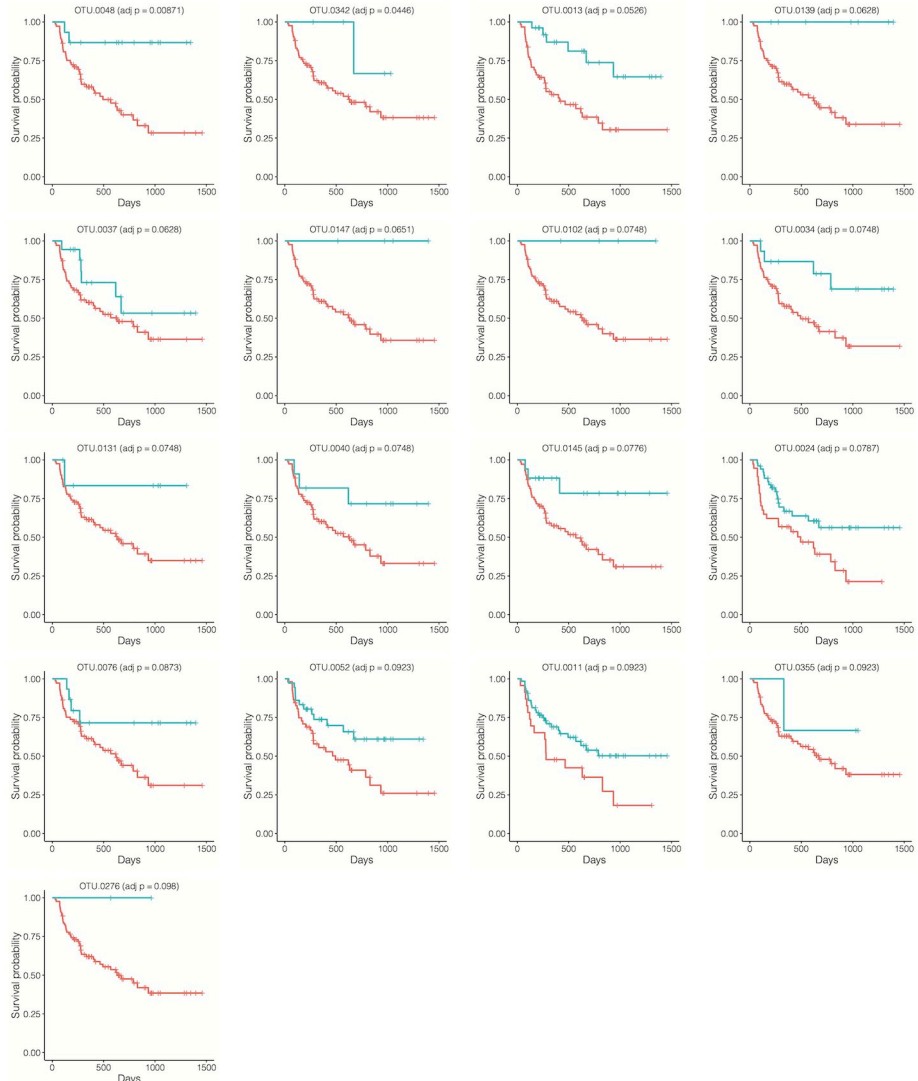

**Fig 4. Survival functions for the overall survival outcome by the presence (blue) and absence (red) status (based on a singly rarefied OTU table) of the OTUs detected by LDM-c.** The plots were ordered by the adjusted *p*-values from LDM-c.

model as continuous covariates. Unlike existing methods which only give community-level (global) tests, our extension of the LDM gives both community-level and taxon-level association tests. Further, we find that the LDM global test and permanovaFL outperform the existing permutation-based global tests, MiRKAT-S and OMiSA, when there are strong confounders.

Although the analysis of a single type of residuals can make use of existing code of the LDM or permanovaFL, the test that combines the two, which is recommended over each single test, does entail additional programming and has been added to the LDM package. Note that the only additional computational burden for testing survival outcomes in the LDM framework is the single calculation of the Cox model residuals and the calculation of the combination test, which is a negligible addition in computation.

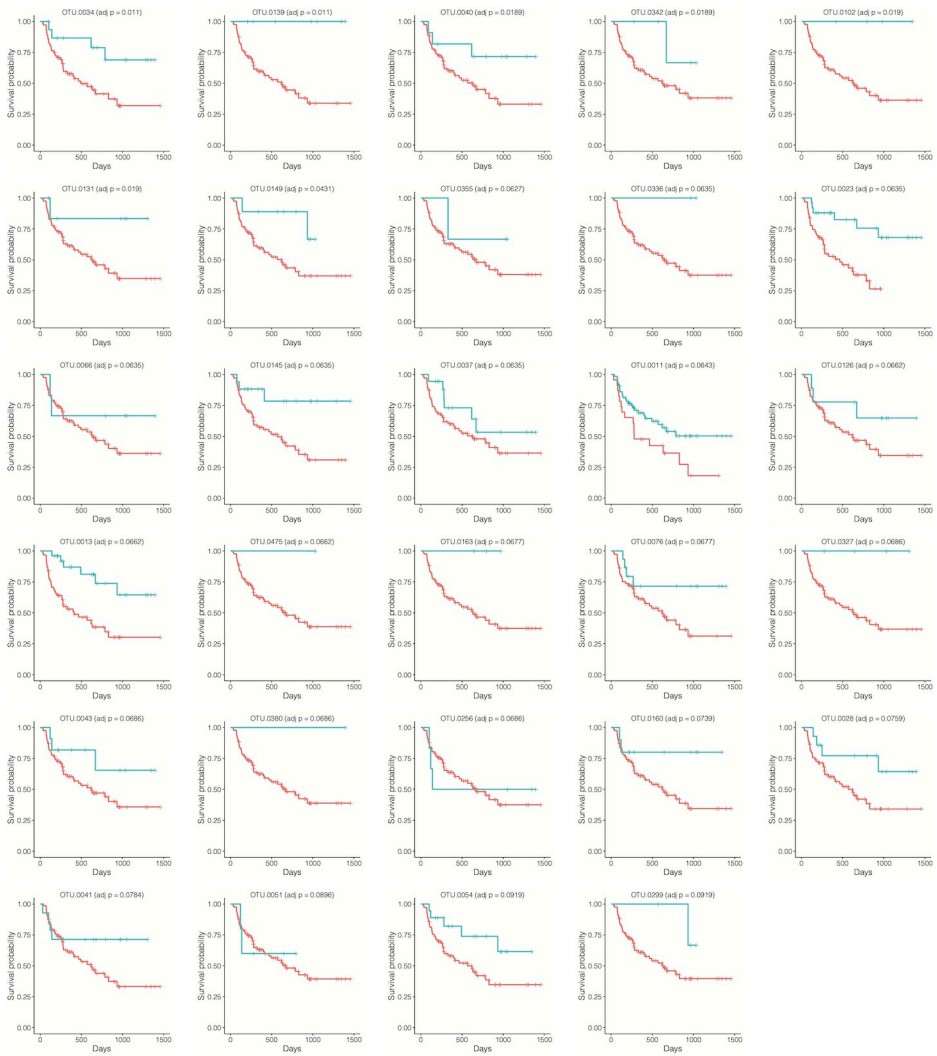

**Fig 5. See the caption to Fig 4.** The outcome is the time to stage-III aGVHD here.

The gut microbiome data in the aGVHD dataset that we have analyzed here were generated from 16S rRNA sequencing. Our approach is readily applicable to microbiome data generated from shotgun metagenomic sequencing, although these data have different error profiles than 16S rRNA sequencing data. In fact, in a recent publication [28], we have applied permanovaFL using the approach developed here to analyze the shotgun metagenomic sequencing data of the gut microbiome and the outcome data on progression-free survival that were generated from a melanoma immunotherapy study [29].

In our simulation studies where the data were generated from the Cox model, we found that the tests based on the Cox model made many discoveries including excessive false discoveries, leading to inflated FDR. Conversely, in our analysis of the aGVHD data, we found that the Cox model made fewer discoveries and particularly zero discoveries based on the relative abundance data. This disagreement reflects the fact that the aGVHD data do not follow the Cox model exactly. Indeed, some survival functions in Figs 4 and 5 showed violation of the proportional hazards assumption.

In this article, we have primarily considered testing hypotheses that are expressed in terms of relative abundances. Some investigators may prefer to test hypotheses that are expressed in terms of ratios of counts or relative abundances. To this end, a common approach is to normalize the read count data using methods such as GMPR [30] and CSS [31] and then apply tests of differential abundance (such as the LDM) to the normalized data. This approach critically depends on the validity of the normalization method of choice, which may not perform well in the presence of sparse read count data. In addition, the LDM can be directly applied to log-ratio data, although some of the appealing features of the LDM such as analyses on multiple scales must be suppressed.

## Supporting information

**S1 Text. Supplementary text.**
(PDF)

**S1 Table. Type I error of the global tests for simulated data in other cases.**
(PDF)

**S1 Fig. Results in scenarios M3 and M4 when taxa 11 and 21, respectively, were associated with the event time.** Results of sensitivity and empirical FDR were obtained when $X_i$ was a confounder ($\beta_{XZ} = 0.8$).
(PDF)

**S2 Fig. Results in the scenario when rare taxa (taxa 91–100) were associated with the event time.** Left column: data were simulated and analyzed based on the relative abundance scale, same as in model M1. Right column: data were simulated and analyzed based on the presence-absence scale (except for OMiSA), same as in model M2. The censoring rate was 50% and $n = 100$. Results of sensitivity and empirical FDR were obtained when $X_i$ was a confounder ($\beta_{XZ} = 0.8$).
(PDF)

**S3 Fig. Results for simulated data with 75% censoring and $n = 100$.** Results of sensitivity and empirical FDR were obtained when $X_i$ was a confounder ($\beta_{XZ} = 0.8$).
(PDF)

**S4 Fig. Results for simulated data with 25% censoring and $n = 100$.** Results of sensitivity and empirical FDR were obtained when $X_i$ was a confounder ($\beta_{XZ} = 0.8$).
(PDF)

**S5 Fig. Results for simulated data with 50% censoring and $n = 50$.** Results of sensitivity and empirical FDR were obtained when $X_i$ was a confounder ($\beta_{XZ} = 0.8$).
(PDF)

**S6 Fig. Sensitivity and empirical FDR of the taxon-specific tests in analysis of simulated data with a confounder $X_i$ ($\beta_{XZ} = 0.8$), 50% censoring, and $n = 100$.** The overdispersion parameter ("disp") varied from 0.02, 0.002, to 0.0002. The results with overdispersion 0.02 are the same as those in Fig 1 (left column).
(PDF)

**S7 Fig. Martingale and deviance residuals, generated from the Cox model that fit age and gender as covariates in analysis of the aGVHD data.**
(PDF)

## Author Contributions

**Conceptualization:** Yi-Juan Hu.

**Formal analysis:** Yingtian Hu, Yi-Juan Hu.

**Funding acquisition:** Yi-Juan Hu.

**Methodology:** Yingtian Hu, Yunxiao Li, Glen A. Satten, Yi-Juan Hu.

**Project administration:** Yi-Juan Hu.

**Software:** Yingtian Hu, Yunxiao Li, Yi-Juan Hu.

**Supervision:** Yi-Juan Hu.

**Writing – original draft:** Glen A. Satten, Yi-Juan Hu.

**Writing – review & editing:** Yingtian Hu, Glen A. Satten, Yi-Juan Hu.

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
