## [Decision Letter · Decision Letter 0]

15 Jun 2022

Dear Dr. Hu,

Thank you very much for submitting your manuscript "Testing microbiome associations with survival times at both the community and individual taxon levels" for consideration at PLOS Computational Biology.

As with all papers reviewed by the journal, your manuscript was reviewed by members of the editorial board and by several independent reviewers. In light of the reviews (below this email), we would like to invite the resubmission of a significantly-revised version that takes into account the reviewers' comments.

We cannot make any decision about publication until we have seen the revised manuscript and your response to the reviewers' comments. Your revised manuscript is also likely to be sent to reviewers for further evaluation.

Sincerely,

Dina Schneidman-Duhovny

Software Editor

PLOS Computational Biology

Reviewer's Responses to Questions

**Comments to the Authors:**

Reviewer #1: In this manuscript, Hu et al. extended their LDM framework to analyze survival outcomes. By treating the Martingale or deviance residuals as a covariate, their LDM framework can be conveniently applied. One nice thing about the proposed method is that it achieves both the community-level and individual taxa-level tests. Simulations and real data analysis demonstrated the superior performance of the proposed method to the ad hoc Cox model (individual taxa tests) and MiKRAT-S and OMiSA (global test). As clinical research has been increasingly adding the microbiome component, statistical methods for integrating the microbiome data into survival analysis are urgently needed. However, previous research mainly focused on community-level tests. Methods for individual taxa-level tests predominantly use the traditional Cox regression model by treating the taxon abundance as a covariate. It is not clear whether this method is valid or efficient. In this work, the authors revealed that the Cox model could not control the type I error properly, which is quite surprising. Overall, I believe the work is a nice contribution to the microbiome community. Once fully validated, I envision it to be an essential tool in statistical analyses of microbiome data. I have a couple of minor questions and comments that need the authors’ feedback.

1. I do not quite understand why the traditional Cox model has such a high FDR inflation. If the null model is generated according to the Cox model, I will expect the Cox model could control the type I error very well.

2. The authors did not mention/address the potential compositional effects. For example, what if several highly abundant taxa are associated with survival in the same direction? I think the authors need to address it in the discussion by providing some possible solutions, e.g., using robust normalization (GMPR, CSS, etc.).

3. In the simulation, the 20% FDR cutoff may be too high. I think the authors should increase the signal strength instead of using a higher FDR since the real data uses 10% FDR.

4. In the discussion, the authors mentioned the LDM-omni3. Could the authors provide the omni3 p-value for the real data?

Reviewer #2: In this manuscript, Hu and colleagues present a method for associating

microbiome with censored survival times. They extend their previous method (and

package, so that the method is available for others) to be able to both detect

community associations as well as associating particular taxa with the outcomes.

Overall, the methods appear to be well done and the benchmarks based on

simulated data do show the advantages of this method.

However, the application to real data shown in the current version is still

limited. I think it would strengthen the manuscript to present more of the

analyses of the aGVHD data in the main text. In particular, I recommend moving

Fig S7 to a main figure. One important claim that the authors make is that

applying the Cox model in an ad hoc model leads to unacceptably high false

discovery rates. This is demonstrated in simulated data, but it would be

important to see the effect in real data as well.

Although not something that I would personally consider to be strictly

required, adding some shotgun metagenomics data analyses would also further

bolster the case that this method is widely applicable. In principle, it should

be possible to apply the method; but shotgun data does have different error

profiles.

**Have the authors made all data and (if applicable) computational code underlying the findings in their manuscript fully available?**

Reviewer #1: Yes

Reviewer #2: Yes

PLOS authors have the option to publish the peer review history of their article (what does this mean?). If published, this will include your full peer review and any attached files.

Reviewer #1: No

Reviewer #2: No
---

## [Decision Letter · Decision Letter 1]

4 Aug 2022

Dear Dr. Hu,

Thank you very much for submitting your manuscript "Testing microbiome associations with survival times at both the community and individual taxon levels" for consideration at PLOS Computational Biology.

As with all papers reviewed by the journal, your manuscript was reviewed by members of the editorial board and by several independent reviewers. In light of the reviews (below this email), we would like to invite the resubmission of a significantly-revised version that takes into account the reviewers' comments.

Please address the point raised by Reviewer 2.

We cannot make any decision about publication until we have seen the revised manuscript and your response to the reviewers' comments. Your revised manuscript is also likely to be sent to reviewers for further evaluation.

Sincerely,

Dina Schneidman-Duhovny

Software Editor

PLOS Computational Biology

Dina Schneidman-Duhovny

Software Editor

PLOS Computational Biology

Reviewer's Responses to Questions

**Comments to the Authors:**

Reviewer #1: The authors have done a great job in addressing my comments. I think the manuscript is in very good shape and ready for publication. Thanks for this important work!

Reviewer #2: I thank the authors for their answers and improvements to the manuscript.

However, unless I am mistaken, the new results in Table 2 (related to the results of the Cox model) actually show the opposite of what was claimed based on simulated data: the Cox model returns fewer positives (in several instances, zero taxa) than the alternatives. Thus, it becomes hard to see how this corresponds to unacceptably high false positive rates (rather than its opposite, a lack of statistical power). I feel that this point should be clarified before publication.

**Have the authors made all data and (if applicable) computational code underlying the findings in their manuscript fully available?**

Reviewer #1: None

Reviewer #2: Yes

PLOS authors have the option to publish the peer review history of their article (what does this mean?). If published, this will include your full peer review and any attached files.

Reviewer #1: **Yes: **Jun Chen

Reviewer #2: No
---

## [Decision Letter · Decision Letter 2]

23 Aug 2022

Dear Dr. Hu,

We are pleased to inform you that your manuscript 'Testing microbiome associations with survival times at both the community and individual taxon levels' has been provisionally accepted for publication in PLOS Computational Biology.

Please update the final version based on the comments of the Reviewer #2.

Best regards,

Dina Schneidman

Software Editor

PLOS Computational Biology

Reviewer's Responses to Questions

**Comments to the Authors:**

Reviewer #2: My concerns have been addressed as the new paragraph explicitly addresses the discrepancy between simulation and real data.

I will defer to the editor as to whether it is also appropriate to make it more explicit earlier that references [10,11] which are used to support the statement that "it is known that small sample sizes and sparse count data may lead to inflated type I error when using the Cox model" are not microbiome-related (which may explain why the results in this manuscript contradict this statement).

The second sentence in the new paragraph should probably read "Conversely, in our analysis of the aGVHD data, we found that the Cox model made fewer discoveries and particularly zero discoveries based on the relative abundance data." ("discoveries" and not "discovery").

**Have the authors made all data and (if applicable) computational code underlying the findings in their manuscript fully available?**

Reviewer #2: Yes

PLOS authors have the option to publish the peer review history of their article (what does this mean?). If published, this will include your full peer review and any attached files.

Reviewer #2: No

---

## [Editor Report · Acceptance letter]

8 Sep 2022

PCOMPBIOL-D-22-00454R2 

Testing microbiome associations with survival times at both the community and individual taxon levels

Dear Dr Hu,

I am pleased to inform you that your manuscript has been formally accepted for publication in PLOS Computational Biology. Your manuscript is now with our production department and you will be notified of the publication date in due course.

With kind regards,

Anita Estes
